# The Journey to Sustainable Participation in Physical Activity for Adolescents Living with Cerebral Palsy

**DOI:** 10.3390/children10091533

**Published:** 2023-09-10

**Authors:** Gaela Kilgour, Ngaire Susan Stott, Michael Steele, Brooke Adair, Amy Hogan, Christine Imms

**Affiliations:** 1Department of Paediatrics, The University of Melbourne, Parkville, Melbourne, VIC 3052, Australia; gaela.kilgour@mcri.edu.au; 2Department of Surgery, The University of Auckland, Grafton Road, Auckland 1023, New Zealand; s.stott@auckland.ac.nz; 3School of Allied Health, Australian Catholic University, 1100 Nudgee Road, Banyo, QLD 4014, Australia; michael.steele@acu.edu.au; 4Grow Strong Children’s Physiotherapy, Melbourne, VIC 3185, Australia; brooke_adair@hotmail.com; 5Cerebral Palsy Society of New Zealand, Auckland 1023, New Zealand; amy@cpsociety.org.nz

**Keywords:** cerebral palsy, sustained participation, physical activity, adolescent, interpretive description

## Abstract

Purpose: To understand adolescents’ and their parents’ perspectives on ‘being active’, this study explored the experience of participation in physical activity (PA), the role of long-term participation in PA, and the importance of remaining active for life. Methods: Eight ambulant adolescents with CP (aged 11–16 years, seven male) participated in a high-level mobility programme twice per week for 12 weeks. Guided using interpretive description, adolescents and 12 of their parents were interviewed before, after and nine months following the programme. Thirty-eight interviews were coded, analysed, and interpreted, informed by audit information, reflective journaling, and team discussions. Results: Adolescents and their parents highly value being active now and into adulthood. Sustainable participation in PA requires adolescents and families to navigate complex environments (interpersonal, organisational, community, and policy). Core themes were: ‘Just Doing it’, ‘Getting the Mix Right’ (right people, right place, right time), ‘Balancing the Continua’ and ‘Navigating the Systems’. The continua involved balancing intra-personal attributes: ‘I will try anything’ through to ‘I will do it if I want to’ and ‘It’s OK to be different’ through to ‘It sucks being disabled’. Conclusions: The journey to sustainable participation was complex and dynamic. Experiences of successful journeys are needed to help adolescents with CP “stay on track” to sustainable participation.

## 1. Introduction

The World Health Organisation’s (WHO) daily physical activity recommendations [1] are that “all adults should undertake 150–300 min of moderate-intensity, or 75–150 min of vigorous-intensity physical activity, or some equivalent combination of moderate-intensity and vigorous-intensity aerobic physical activity, per week. Among children and adolescents, an average of 60 min/day of moderate-to-vigorous intensity aerobic physical activity across the week provides health benefits” (p. 1459). Maintaining participation in physical activities at home, school and in one’s community has many benefits: an improved sense of self, autonomy and engagement, greater physical mastery, meaning and challenge, belongingness, and enhanced physical and mental health [2,3]. Globally, however, it is estimated that the WHO recommendations are met by only three in four adults and between 10–20% of adolescents [4,5]. For adolescents and adults with disabilities, the ability to meet the guidelines is further reduced, and little is known about how to achieve their sustained participation in physical activity [6,7]. Overall, the international uptake of WHO physical activity policies for people with disabilities into practice has been slow [8]. In addition, a major void appears to exist in determining whether physical activity interventions are effective beyond the period in which they are researched, what outcomes are sustainable and, for whom [9,10]. There has been a call for more longitudinal studies and mixed methods research to identify the components and strategies required for sustainable, long-term participation in physical activity for all ages [11,12].

We recently conducted a 13-month, longitudinal study with eight adolescents with cerebral palsy, assessing whether participation in a high-level mobility programme (HLMP) designed to train running skills could lead to increases in physical activity during and after the intervention [13]. We defined participation as attending (being there) and being involved in physical activity (may comprise elements of motivation, persistence, engagement, affect, and possible social connections), consistent with the Family of Participation Related Constructs (fPRC) framework [14]. Quantitative findings were that adolescents had high levels of attendance, involvement and goal attainment during the 12 weeks of the HLMP. Beyond the programme, frequency and duration of participation were variable, goal attainment was highest and most consistent for involvement goals, and physical capacity increased for seven adolescents in at least one running test. Despite these positive changes, a disconnect between goals achieved (physical competence, desired participation attendance and involvement) and participation outcomes (actual attendance and involvement) and physical capacity changes were found. That is, increased physical performance, capacity, and involvement did not always translate into increased attendance in physical activity. Sustainable participation in ongoing physical activity appeared to be challenging, irrespective of physical gains, high levels of involvement and positive intentions. Other researchers have also found that interventions aimed to improve physical activity outcomes have infrequently translated into increased ongoing participation in physical activity [15,16].

Understanding what is needed to assist adolescents with cerebral palsy to be active now and in the future may help to determine strategies that continue beyond the period of the physical activity intervention to be embedded into every day home and community participation. Implementation of such strategies may move adolescents with cerebral palsy from high levels of inactivity to habitual activity. However, research addressing experiences longitudinally before, during, and after an intervention to increase physical activity is lacking, with most knowledge gleaned from interviews of children and adolescents with disabilities as a ‘snap shot’ during or at the end of an intervention programme [17,18,19,20,21,22] or from surveys [23].

To develop a greater depth of understanding using a longitudinal approach, we conducted an interpretive description qualitative study over the 13-month study period to explore the journey to sustainable participation in physical activity for the eight adolescent participants with cerebral palsy (CP) and their parents. Interpretive description was developed originally for nursing and is a methodology used to capture experiences and the ‘whole story’ [24]. The clinical base for interpretive description has expanded to include experiences of physical activity in young girls and adolescents [25], barriers and facilitators to physical activity for children with cerebral palsy in specialist schools [26], and the experiences of players and parents when participating in power soccer sport [17]. Using this interpretive method, novel strategies and practice changes can be identified that may not have been known, or considered, without the voices of the participants [17].

### 1.1. Aims

The aims were to (1) explore the experience of participation in physical activity and the importance of remaining active for life from the perspectives of adolescents with cerebral palsy who took part in an HLMP and their parent/s and (2) gain a greater understanding of how to encourage sustained participation in physical activity in adolescents and their families in everyday life, and (3) inform practice, research, and most importantly, the adolescents with cerebral palsy and their parents.

### 1.2. Research Question

What could adolescents with cerebral palsy and their parents tell us about the value of participation in physical activity and sustaining physical activity into adulthood?

## 2. Material and Methods

### 2.1. Design

This interpretive description study was completed as part of a larger, non-randomised, concurrent, single-subject research design (SSRD) study investigating the effect of an HLMP on participation in physical activity [13]. The study was concurrent to, but not mixed, with, the SSRD. The study was approved by the Health and Disability Ethics Committee of New Zealand (19/STH/22) and registered with the Australian New Zealand Clinical Trials Registry (trial identification number ACTRN 12619000126112; universal trial number U1111-1226-8425). The HLMP was a community-based intervention focused on task-specific training of running skills for one hour, twice per week, for 12 weeks. The primary outcomes of the SSRD were attendance and involvement in physical activities, and secondary outcomes were changes in physical competency over a 13-month period.

Interpretive description was chosen to address the clinical questions posed about long-term participation in physical activity, with the aim of informing clinical and community practice and creating change [24,27]. The process was inductive, informed by the experiences and realities of children with cerebral palsy and their parents prior to, during and nine months following the HLMP, and their thoughts for the future.

### 2.2. Child and Parent Characteristics

Children aged 7–18 years with a diagnosis of cerebral palsy classified at Gross Motor Function Classification System (GMFCS) level I–II [28] were sought via purposive sampling from health care and community settings to participate in the SSRD [13]. Participants able to follow instructions, set goals and express their ideas and experiences when interviewed were included. Parent demographic data were not collected. All SSRD participants and a parent of each (one or both, as chosen by the family) were invited to participate in the qualitative component and were required to speak English. Each participant provided informed, written consent.

### 2.3. Researcher Characteristics

The lead researcher, GK, an experienced physical educator, coach, and physiotherapist, took on a ‘learner role’ to minimise any unintended biases during sampling, data collection, and analysis [29,30]. GK’s personal experiences, knowledge and understanding of the phenomenon were valued in interpretive description. The diversity and experience of the research team in the areas of participation, qualitative methods, cerebral palsy and physical activity contributed to the interpretation. The team included an occupational therapist (CI), a physiotherapist (BA), an orthopaedic surgeon (NSS), a biostatistician (MS), and a young woman with lived experience of cerebral palsy (AH).

### 2.4. Data Collection

Qualitative data included three semi-structured interviews, an audit trail, reflective journaling, dialogue with the team and others, and member checking. The semi-structured interviews were conducted at baseline, following the 12-week HLMP, and at 9 months post-completion of the HLMP. The baseline semi-structured interviews were piloted with nine children (aged 7–16 years), with changes made using feedback prior to implementation. The following two interview schedules were developed as the HLMP progressed, based on participant and team member questions and reflections. Interview schedules two and three were also piloted prior to delivery, with the same nine children as in interview one, to ensure age-appropriate language and meaning. The questions explored the experiences and perspectives of both the children with cerebral palsy and parents of sustained participation. For example, the child’s interview explored: *Can you tell me what has helped you to be active and keep being active after the HLMP? And what will be needed in the future*? For the parents, the focus was: *Can you tell me what is needed to help your child to be physically active and to develop a habit of physical activity over time*? *What do you think will be needed into adulthood?*

All interviews were conducted by GK and audio-recorded. The questions were emailed to participants 2 weeks prior to support preparation and potentially greater depth of response in the interview. The child–parent dyad chose the interview location, date, and time, and interviews were not time limited. The child–parent dyad could choose whether to attend interviews together, and the same dyad was encouraged to attend all three interviews. All interviews were conducted face-to-face in a community setting, except for one goal-setting session that took place using video-conferencing technology at the child’s request. Interview style and delivery were guided by Listening to the voices of children with disabilities in New Zealand—Office for Disability Issues (https://www.odi.govt.nz/guidance-and-resources/listening-to-the-voices-of-children-with-disabilities-in-new-zealand/, accessed on 12 March 2019). Field notes and reflective thoughts were taken during the interviews.

### 2.5. Data Management

All interviews were transcribed verbatim using Otter (Otter.ai, Los Altos, CA, USA) and tabulated in Excel (Microsoft Corporation, Redmond, WA, USA). Data coding was undertaken in Excel to link the respondent to the narrative, the meaningful statement, preliminary code, and preliminary concepts/patterns/themes. Voice recordings and transcriptions were stored securely as per ethical requirements.

### 2.6. Analysis

Transcripts were re-read throughout the analysis by GK to ensure each child’s and parent(s)’ story was reflected on and meaning was not lost as analyses progressed. Meaningful statements were extracted from the first interview, and preliminary codes were developed to help plan Interview 2 and 3 schedules. Full analysis and interpretation occurred concurrently at the end of the third interview using the codes to inform the analysis [27]. All coding and thematic analyses were completed by GK, with all six authors reading and listening to two-three transcripts to assist with reviewing and interpreting the developing codes throughout the process. The final interpretation confirmed the overall themes and resulting subthemes, which were summarised.

### 2.7. Strategies for Enhancing Rigour and Trustworthiness

Rigour and reflexivity were addressed by acknowledging the inherent biases of GK when delivering the HLMP, interviewing, conducting and analysing the results of the HLMP [27]. The potential biases were managed by including interviews, journaling, an audit trial and open discussions among the authors and with people outside of the study. Each child and parent were invited to review, critique and provide feedback on the concepts, subthemes and themes. The member-checking processes resulted in the development of the metaphor for the study: the journey of sustained participation and the use of the waka. Transparency and trustworthiness of data collection, analysis and interpretation were developed using two checklists: the Standard of Reporting Qualitative Research (SRQR) [31] and the Consolidated Criteria for Reporting Qualitative Studies (COREQ) [32].

## 3. Results

### 3.1. Participants

Eight adolescents (mean age 13 years 11 months, SD 1 year 6 months, seven males) participated in both the SSRD and the Interpretive Description. The term adolescent is used hereafter as representative of the participants’ age range of 11–16 years [4]. All adolescents mobilised independently without aids at GMFCS levels I–II [28]; six attended mainstream schooling, and two attended supported learning units. Five of the adolescents were involved in at least one formal, organised physical activity prior to the programme, e.g., karate, athletics, swimming lessons, cricket, and football. No adolescent received any form of physiotherapy during the intervention or follow up period. A median of 23/24 HLMP sessions were attended by the participants. The one female participant only attended two sessions but completed all follow up testing and interviews

Eight mothers and four fathers participated in a total of 38 semi-structured interviews, which took between 15–45 min. The interview schedule can be found in Appendix A. The adolescent-parent dyad was self-selected based on those who attended the HLMP most frequently and remained consistent throughout. Four adolescents had both parents attend at least one interview, and three chose to be interviewed independently. Parents provided support during interviews for their adolescents who communicated at Communication Functional Classification System level II [33] to assist with slower communication. Following interviews, additional information was provided from the adolescent-parent dyad via email, telephone or text, e.g., further reflections or specific examples of their interview responses that they had not considered during the interview. Full data collection occurred for seven dyads, with one dyad leaving the country prior to their 9-month follow-up interview.

In reporting, quotes from adolescents are denoted with (A) and parents with (P). Identifiers are otherwise not included to reduce the likelihood of identifying participants.

### 3.2. Themes and Synthesis of Findings

The interpretation of the findings resulted in an organising structure entitled ‘Journey of Sustained Participation’ that conceptualised the themes and subthemes. The concept of ‘Journey’ refers to the ideal of moving forward to achieve ongoing, lifelong participation in physical activity and acknowledges the reality that the ‘Journey of Sustained Participation’ describes the complexity of sustaining participation via the dynamic interplay of the themes and subthemes. There were four themes: (i) Just doing it, (ii) Getting the mix right, (iii) Balancing the continua, and (iv) Navigating the systems (see Table 1 and Table 2).

### 3.3. Themes

#### 3.3.1. Just Doing It

‘Just doing it’ describes the parents’ experiences of getting their adolescent to ‘have a go’ and the adolescents’ perspectives that to start being active requires ‘just doing it’. As well as attending, the adolescent needs to be involved in the activity to move towards being ‘on track’ to sustained participation.

There was some variation in the perceived frequency, diversity, and duration of ‘just doing it’ and requirements to sustain current and future participation. Some parents reported getting their adolescent ‘out the door’ was challenging and required ‘pushing and forcing’ at times. Adolescents and their parents valued diversity in activities, but this did not always translate to increased or sustained frequency or duration:

*I think, like with most people, you get sick of something so you need a break from it. So that’s probably quite important, variation*.(P)

Many factors were identified as limiting attendance, including parents’ time, own health and other family pressures, adolescents’ lack of motivation or change in preferences, and a feeling that with increasing age, it would reduce naturally:

*I don’t think it’s a reflection of the programme. Like his physical activity has dropped. However, that’s just been a combination of his age and COVID and starting high school and just a timing thing*.(P)

Sustained participation in physical activity was more complicated than being able to attend the activity. Both adolescents and parents reported that ‘being involved’ was a prerequisite to physical activity being sustained now and into the future. Adolescents found that the benefits of being involved included positive affect and well-being:

*It distracted me. With my anxiety. They’ve [the gym instructors] helped me straighten my mind for about, you know, half an hour or 45 min of the programme*.(A)

Choosing to be involved in physical activity was not always easy, and not all adolescents had intrinsic motivation to be active:

*Just going to play with a friend. Gets him out. Just to see if there’s an activity at the end of it that he looks forward to. However, mostly just ice creams and things…… You know he doesn’t get a high from being physically exhausted. He gets a high from the, the rest afterwards and whatever it is he does afterwards*.(P)

‘Just doing it’ reflected the adolescent’s journey could be effortful, challenging and not always fun due to the ‘in the moment’ personalised nature of involvement. The role of the experience of ‘involvement’ in physical activity has been described in detail elsewhere [34].

#### 3.3.2. Getting the Mix Right

‘Getting the mix right’ involves the ‘right people’, ‘right time’, and ‘right place’ for an adolescent with cerebral palsy to stay on track to sustain participation in physical activity.

Right people. Adolescents and their parents refer to people who motivate, advocate, encourage, and support the adolescent to be active as the ‘right people’. The ‘right people’ were diverse and could be members of the family, the school, organisations, or within the broader community. Adolescents consistently acknowledged their parents, and to a lesser extent siblings, as essential to their journey:

*My mum mostly encouraged me to do the biking. My brother and my dad have done frisbee a few times, so I decided to join them because dad wanted me to*.(A)

His father agreed:

*I think it was the family, us going biking, is what got him started. Got his confidence up. Just got him over the hurdle*.(P)

Parents’ role as the ‘right people’ to keep their adolescent active was multi-faceted:


*Transportation to places, making time available, money for ice chocolates, keeping them on task, reminding him. Just helping him be organised to be ready to go, having a water bottle, nutrition…*
(P)

Being active with friends was critical for some, but not all, adolescents. Friends were valued for company, competition, support, and comparison:

*I find it difficult to do anything on my own. Having people by my side to not only support but to have fun with. If I make mistakes, and there’s no one to pick me up and support me, it’s a bit more difficult, because I get disheartened*.(A)

Participating in community programmes, sports, and within organisations can also be influenced by the ‘right people’, with the social aspect of physical activity and the sense of belonging an essential requirement for some adolescents to start or continue to participate:

*It’s about the people who do it with him… they make it about the social. It’s not about the slog up the hill, it’s about everyone getting up there*.(P)

Right time. For some adolescents, the ‘right time’ related to maturity, time management skills, and ability to take on self-responsibility for being active and ‘getting out the door’:

*I’ll organise what time suits everyone. I’d have to ask everyone what days’ work and if the little kids [siblings] have got something [that has to be done]… probably need to get in the habit of going for runs on certain days or certain times*.(A)

Some adolescents also stated there might be a ‘time limit’ for some activities, such as hydro sliding or gym classes, based on age. The ‘right time’ was also described as ‘their time’ to participate in an activity of choice with a specific parent (e.g., squash with dad) or without interference from others:

*I prefer if it’s just me by myself. I just feel like I need some time away from my brother or some people that have been frustrating me, sometimes at school and stuff*.(A)

The ‘right time’ for the family unit was complex and included consideration of timing of the activity (e.g., day(s) of the week, length of sessions, travel time, time within a season), adolescents being allowed to do one activity only, enough time for school work, parents managing being ‘time-poor’ and their own schedules, and negotiating finances/funding to offer their adolescents an experience at the ‘right time’. Planning family physical activity at the ‘right time’ was important so no one ‘pulled the pin’ (P). Timing is also related to managing multiple appointments for an adolescent with a disability. Balancing ‘right time’ with the needs of siblings was often challenging, and the balance may or may not go in favour of the adolescent with cerebral palsy:

*We tend to revolve around a lot of what his sister has to do, because she needs a lot of transport into town, so that affects a lot, [A] would miss out quite a bit*.(P)

In contrast, one parent said:

*… because everything’s so much harder and more of an effort for him that actually you almost overcompensate a little bit, and you sort of run rings around him just to ensure he’s getting those things*.(P)

Right place. ‘Right place’ was identified as being where support was available for the adolescent to achieve their goals, try new activities, have positive experiences of physical activity, meet people, and feel heard. Adolescents ‘expected to have fun’ at the ‘right place’ and ‘feel great’ after being active. The ‘right place’ also included environments with accessible and affordable resources, programmes and instructors to ensure physical activity could occur and be sustained.

Planning ahead to remove potential hurdles in each ‘place’ was considered essential to ensure sustained participation. Both parents and their adolescents reported poor experiences at school. Schools should be the ‘right place’ with the ‘right people’ for students to experience success with their peers. Adolescents talked about their role in advocating for themselves at school:

*I see this in PE. People don’t actually think with their brains. How is this going to work for somebody that has one side of their body not working or has major weakness in the side of the body?… I said to my teacher ‘If I’m not able to do it, are you able to come up with something I’m actually able to do to the best of my ability, not only as a student in your class, but as somebody that actually values sport within life’. He [the teacher] was very slow in the start, but I helped him figure it out in a way that he understood what I had struggles with… because he knows I like to be involved. It was the teacher opening up to seeing how adaption can actually be good for me*.(A)

Negative disabling experiences with community sports in the past were also reported:

*He didn’t enjoy [club name] athletics attitude towards paras. I think he got broken by turning up every week and being last*.(P)

Both parents and adolescents wanted their communities to provide the ‘right place’ to be active. However, choices were limited, barriers to accessibility were common, and the ‘getting the mix right’ was hard to achieve:

*If things don’t work out at high school or sports, I’ll have to create something… The reason that he’s chosen to be inactive is because, you know, barriers have come up, not out of choice… I think he would prefer to do an activity than not at all*.(P)

#### 3.3.3. Balancing the Continua

‘Balancing the continua’ refers to the ever-changing factors that influence whether an adolescent chooses to be active in the moment and over time. Table 2 contains quotations from adolescents and their parents that highlight the complexity of ‘balancing the continua’ and describe the subthemes and components that determine which direction the adolescents moved along the continua, i.e., from the positive end of being active and sustaining participation (‘I will try anything’ and ‘It’s OK to be different’) to the negative end where participation slows or stops, and the adolescent may become inactive (‘I will do it if I want to’ and ‘It sucks being disabled’).

‘Balancing the continua’ refers to the need to readjust the positioning each time an adolescent chooses to be active and when active, with the continua of some adolescents being fragile and easy to unbalance and others being stable and robust. ‘Balancing the continua’ could be rewarding, self-assuring, and motivating or could shift to being demanding, never-ending, and potentially exhausting, both physically and mentally.

#### 3.3.4. Navigating the Systems

Adolescents’ and parents’ experiences of participation in physical activity did not always meet their expectations and was ‘a minefield’ of systems to navigate. This minefield included a complex dynamic interplay between a variety of people (e.g., parents, friends, coaches), organisations (e.g., school, therapy), community groups (e.g., physical activity programmes, including the HLMP), and government policy decisions (e.g., the COVID pandemic), all of which interacted at various times and at various levels of intensity during a child’s journey to becoming active.

‘Navigating the systems’ emerged as a separate theme but also interacted with and influenced other themes. Parents reported liaising with other adolescents with disability and their parents as the ‘right people’ who had already navigated the systems. These ‘right people’ could recommend activities, facilitate networking, and provide further knowledge so that they were ‘hearing about other activities, being exposed, sometimes the idea [is] being planted’ (P). Finding the ‘right people’ in education and health and within organisations promoting physical activity was considered essential. ‘Getting the mix right’ and successful ‘navigation of the systems’ may not occur without experienced people. The following dyad talked about a time during a community adaptive gymnastics program when the ‘right people’ were not available:

*She keeps pushing and relentlessly pushing me… It was bit more horrendous last time that happened. I couldn’t do it*.(A)

*She wouldn’t adapt it. It took you four days to get over it*.(P)

Parents talked about their regret and disappointment that opportunities may have been missed due to lack of service provision at the right time:

*If you think there’d been some sort of active, ongoing programme or telling us exactly what to do then, or how to do it, life might have been a bit easier really… A lot of families said if there was something dedicated to these kids around their movement and the running, years and years before, other than kind of what we got, instead “this is as good as it’s gonna (sic) get”*.(P)

Parents reported needing to navigate every ‘place’ that their adolescent was active in so they can ‘try and do the best they can’. Navigating systems was highlighted as complex, frustrating, and exhausting.

Participating in physical activity within government-specified restrictions during New Zealand’s COVID lockdown period was a policy-level decision that adolescents and their families had to navigate. For most adolescents, the policy provided a significant loss to participation opportunities, while others found new opportunities.

#### 3.3.5. Planning for Sustained Participation in Physical Activity

Adolescents and their parents described the importance of being active for life and highly valued physical activity. However, their plans for how to successfully stay ‘On Track to Sustained Participation’ were not always in agreement, required significant planning, and the plans themselves may lack long-term sustainability (Table 3). Adolescents recognised that strategies were needed, especially when they were not intrinsically motivated, and suggested:

*Have a deadline, where I had to do it and have a consequence, or a reward if I did it before a certain time. If I did meet the deadline something good, or if I didn’t something bad would happen*.(A)


children-10-01533-t003_Table 3Table 3Planning for Sustained Participation in Physical Activity.PlansSample of Supporting Quotes
**
*Adolescent strategies*
**
Right peopleSetting deadlinesPurposeful choices/preferencesFriends (not essential and less into adulthood)
*Having people to do it with, having activities I like, having a reason to do that and having a reason to keep doing it…At least something five times a week. At least. Cuz (because) two days off is almost too much.* (A)*Just keeping an eye out for activities at school, extracurricular activities, getting involved with things my friends like so joining maybe a club my friends are in*. (A)
**
*Parents Strategies*
**
Structured programmeConstant encouragementRight peopleBeing role modelsNovel activitiesFun, purposeful and varied activitiesSupport preferencesExtrinsic rewardsHaving enough time and being fit enough themselves
*Structured programme helps because you have a time and you have to go*. (P)*Encouragement and doing more of it. If we do more of it, they know that it’s fun’*. (P)*Just to get up and do it. Don’t say you’re gonna (going to) do it as a parent and not do it. Just get up and do it*. (P)*Ice creams, fizz, chocolate and things*. (P)
**
*Set goals and follow dreams*
**
MotivatingGive purpose, meaningfulHelp future planning/timingNew opportunities and options
*Being able to bike opens up my life for future 10, 20, 30 years. If I cannot afford a car and I’m living close to the supermarket, I bike and that would be very beneficial. I may be able to go to the movies with friends cuz (because) that’s not too far and its easy roads*. (A)*I reckon I’ll be around 18, 19, 20 [years old] when I get my black belt (in karate), so that’ll keep me busy for a few more years. I’d definitely like to keep swimming after para swimming. I’ll probably go down to the pool every so often. I’d definitely like to join the squash club, that would be really, really good. I’d start biking places. So, if I go to university, I’d bike to university and back every day, that’d keep me active*. (A) *To eventually represent New Zealand in an international competition which can either be the Paralympics, the World Champs or the Commonwealth Games. I want it to be in the throwing disciplines. However, with a lot of the people from around New Zealand being quite good, I need to push myself to get better than them*. (A)*I did have some fanciful dream about going to the gym, getting stronger, because that would be quite good. It’s mainly just a fanciful idea because maybe if I get active, I won’t need it*. (A)
**
*Value supporters*
**
Parents were the number one supporter’s but often felt undervaluedGrandparents and siblingsRight people willing to learn/experience valued but not essentialAdvocates for inclusion
*Mum helps a lot*. (A)*I think it was the family, us going biking, is what got him started. Got his confidence up. Just got him over the hurdle’*. (P)*I feel like I do give him that support. It gets tiring, constantly nagging and getting nothing for it. I think, if together, we come up with a plan of where it fits in his week, and I help find him a spot then I think he should be able to do that by himself*. (P) 
**
*Build and expand support networks*
**
Peer support in mainstream and disability specific activitiesActive role modelsSocial support outside in the communityRight places—facilities, equipment, resourcesRight people—seek expertise when requiredFinancial commitment to participate in preferred activities
*I think if he’s around people that are active, he’ll join in. Because he’s quite social. That will have an impact on him*. (P)*Trying to include other people that he likes, finding coaches or support people that are knowledgeable and fun and … know a bit about physical and learning abilities’*. (P)*With adult support or peer support as he gets older, having things in place where people are, where those things are organised in advance, that he belongs to a group, if we can and that he has friends or peers that will be involved with him. And using funding as well for that sometimes*. (P)*If he wanted to choose an activity, then we’d be very keen for him to do that. And if we really disliked it or something, then we would try and get somebody else to support him to do it with or we’d just start doing it if it wasn’t our preference. Yeah, until he was able to do it himself*. (P) 


Friends were needed as part of the strategy for some adolescents for ‘a good laugh, for spirit’ but were not seen as important into adulthood and not essential for everyone. Goals were deemed valuable and important strategies to help participation and to direct sustained participation. One parent said that, ‘Goals are a good motivator. We’ve just got to remind him of what his goals are, then he generally makes the right choices’ (P). Goals needed to be timely, meaningful, and updated regularly to help with ongoing motivation.

Parents reported that they felt their role in encouraging activity was essential now and into the future for their adolescents. Parental strategies were varied, encouraging their adolescent to ‘want to do it’ by ‘making it happen’. However, parents had less influence as their children grew older and often felt undervalued and underappreciated:

*He’s not interested in what I suggest, so my pulling-power on those things or my ability to make it fun is not as much as other people*.(P)

Parents were concerned about their ability to help their adolescents continue to be active as adults, in particular, continue to be mobile and healthy, but most lacked specific, targeted strategies to address this:

*I don’t know the answer to getting him to do more outdoor exercise type activities. I don’t have any strategies or plans or thoughts on how to do that yet. I’d have to try and think of ways to inspire him to be more active*.(P)

Some adolescents and parents have experienced the need to be strong advocates for inclusion and opportunities. One parent reported ‘it may not be our choice’ regarding opportunities to stay in mainstream sports or continue with a preferred activity such as cricket for her son. Advocacy by community and organisations has been explored by some parents and experienced both positively and negatively.

As their adolescents planned for greater independence with age and increased the outside influences in their lives, parents described how extensive work was needed to build on current opportunities and for new successful opportunities to be developed. Planning was greatest by parents of adolescents with additional learning needs who would find being independently active more challenging. Handing over the responsibility to the adolescent was a common thread in discussions with parents in terms of their roles in educating and supporting the transition to independent self-management:

*That he’s aware of his abilities and has the motivation to do what he needs to do. For nothing to be a problem for him. It has to come from him eventually*.(P)

Ultimately, parents conceded that it would be their adolescent’s decision to participate or not.

#### 3.3.6. Interpretive Description: Translating the Journey

Interpretation of the themes resulted in the creation of a metaphor using a waka (Figure 1). A waka is a traditional Māori watercraft, like a canoe, used by Māori (indigenous people of Aotearoa, New Zealand) to journey across the Pacific Ocean to discover Aotearoa, New Zealand. Waka are wooden, ranging in size and specification (single and double hull). The ornamental carvings adorning the waka are of deep cultural significance. The waka is ecologically significant in this study as the waka has helped guide and navigate people to places across their lifetime for centuries and is celebrated in a sport called waka ama, promoting participation in physical activity.

The metaphor of an adolescent entering their waka and choosing to journey towards sustained participation in physical activity was developed. For the adolescent to be active and sustain their physical activity, they must enter their waka, choosing to attend and be involved: ‘Just doing it’. The journey can be enhanced or hindered by the mix of people, time, and place in their waka ‘Getting the mix right’, the ability to balance their waka (‘Balancing the continua’) and the ability to navigate the many levels of systems that influence the journey ‘Navigating the systems’.

The aim of each journey in the waka was to stay on track to achieve their goals. The Māori saying ‘He waka kore hoe, He tangata kore huarahi’ (translation: ‘A waka without a paddle is like a person without direction’) indicates the importance of the adolescent using their paddle at the ’right time’ to move their waka forwards and to direct their own journey to the ‘right place’. The journey will be more successful if the ‘right people’ are in the waka: ‘He waka eke noa’ (translation: ‘We are all in this together’). Adolescents will be able to sustain their participation in physical activity if the ‘right people’ are working together (collaborating and connecting), heading to the ‘right place’ at the ‘right time’. A waka will move quickly forward if the adolescent is motivated and confident to “I will try anything/It’s OK to be different” (sails fully out) but will be slowed if self-efficacy, confidence and autonomy are low “I will do it if want to/It sucks being disabled” (sails down/in). The challenge of balancing the continua is reflected in the following quote:

*My son has had a few challenging situations where he refused to do something, and he said to me that he was folding up his sail*.(P)

The risk of being unable to navigate the systems (personal, interpersonal, organisational, community and policy) is the waka may go off course, wobble or tip over, resulting in no participation.

The themes and graphics were developed with families, and the journey for each adolescent was developed collaboratively. The concept of collaboration on the journey was reflected in the following feedback:

*One thing I felt could be included is that while a child is the master of their own waka, it is fundamental that they need to be able to trust their crew to sometimes point the waka in a different direction and show a possibly more interesting path*.(P)

## 4. Discussion

The interpreted description, the ‘Journey to Sustained Participation’ and the metaphor of the waka highlight participation in physical activities as dynamic and complex and the journey as changeable through the life course. The ‘journey’ was seen as challenging yet highly valued by adolescents and their parents. Themes and subthemes described adolescents with cerebral palsy as being able to sustain participation if they were ‘just doing it’, could ‘get the mix right’ by involving the right people at the right time and right place, keep the ‘balance of the continua’ at the positive end by trying anything, feeling OK about being different, and thriving when being active. The journey was not always smooth, and small disturbances could have a large impact, either positively or negatively, on the adolescents and parents. The complexity of ‘navigating the systems’ and the dynamic interplay of factors (family, friends, community, organisational or policy level) has the potential to shift the adolescent’s balance and their waka’s course on their journey of sustained participation. The importance of participation in physical activity is well recognised [17,20,35,36,37,38], and the ‘journey’ must consider an adolescent with cerebral palsy’s real-world contexts and environments to ensure lifelong attendance and involvement [14].

Adolescents who had established activities reported finding their niche (the ‘right place’) and wanting to participate regularly, for example, in a para sports club or a team sport. Unlike in previous reports [39,40,41], adolescents in this study reported being satisfied with their level of attendance, whilst parents reported challenges in ‘getting the mix right’ for attendance. This included the ‘right time’ to be active and ‘balancing the continua’ to avoid ‘I do not want to do it’. Forcing children to participate in physical activity has been reported to be easier in younger children [42] but not so in adolescents. Participation relied heavily on personal preferences (adolescents’ motivation to attend), health-related factors (pain and fatigue), and the social and environmental factors influencing the parent’s ability to ‘make it happen’. If finding the ‘right place’ (or niche) for physical activity is a key driver of sustained participation, then supporting children and families to achieve this is crucial. Being overt about the goals of ‘trying anything’ at various times in a child’s life, drawing on modifiable factors and supporting adolescents when they indicate ‘I do not know if I can’ can be used to tip the continua to the positive end of participation [36]. Targeting modifiable personal factors by working towards positive behavioural change and self-determination may increase attendance and involvement in physical activity [43,44,45,46,47,48].

The ‘Journey to Sustained Participation’ embraces the concept of a journey through life and describes what it takes to sustain physical activity participation needed for a healthy life. Parents reported that they promoted physical activities that were purposeful, safe, liked by their children, good for their health, in supportive environments and fitted in with the family’s schedule [49,50]. Adolescents reported feeling safe, understood, and supported by their families. ‘I will try it if…’ and ‘I will try it but…’ were situated at the positive end of the continua, but both came with provisos: if the new activity was not suitable, enjoyable, or the ‘right people’ were not present, they would not continue [43,51]. All parents faced times when their adolescents asserted themselves and made choices they did not agree with. For parents of adolescents with cerebral palsy, the conflict was compounded by their need to consider the implications of the adolescents’ decisions in terms of their future health. The parents who participated in this research also valued physical activity as a way to help move their adolescent ‘towards normal’, i.e., closer to their peers, to prevent long-term health and functional problems. The parental focus on benefits of participation for body function and structural outcomes has been reported previously [21,52,53,54].

Consistent with Shields and Synnot’s findings [55], adolescents valued participation most when they chose their own activities (‘I like it, and it’s my choice’). Preferred activities tended to be formal, structured, and goal-directed. Adolescents were disappointed when environmental or social obstacles, such as lack of transport, parent’s work commitments, or a clash with a sibling’s activities, prevented attendance. The subtheme of ‘Do I have a choice?’ was based on adolescents and their parents reporting that participation was not always in preferred physical activities, the child’s choice, or enjoyable., e.g., walking the dog and walking to the school bus served a purpose, consistent with previous studies [56,57,58,59]. Clinicians have also found a reduced likelihood of participation if the activity was the parent’s choice rather than the child’s [60]. Unfortunately, children with disabilities are less likely to participate in their preferred activities [59,61,62] despite the potential benefits of developing sustained participation. The question of ‘Whose choice is it?’ remains.

Past research suggests that children with disabilities want to participate in more physical activities and be more active [39,40,41] and that participation in physical activity in early childhood predicts a physically active lifestyle in adulthood [63,64]. This implies that establishing habits very early in life is key [63], given physical activity participation starts to decline in children with cerebral palsy as early as 4 years of age and continues at a low level as they grow older [63,65,66,67,68]. Transition time points, e.g., to high school and subsequently to employment, are also opportunities for loss or gain in participation [63]. Although an individualised approach may be needed for each adolescent, participation in physical activity, starting with ‘Just doing it’, is essential to improve health outcomes for all people with disabilities. Targeting physical activity participation at these transition points might be essential to address the need for sustained physical activity at times of changing priorities and preferences for specific activity situations [22].

Planning for sustained participation requires listening to both adolescents and parents because self-reported frequency, desire to change, and involvement in activities in the home, community, and school have been found to differ between children with disabilities and their parents [69]. Children with disabilities also have a different concept of participation from their parents, teachers, and professionals [70]. Family functioning must be considered as having a significant effect on participation involvement in the home, community and desire for change [71]. Parents in this study identified the need for support and knowledge from local providers, meaning investment must be allocated to training and increasing knowledge in the physical activity sector to support people with disabilities to achieve lifelong participation. Parents are in a unique position to help (or not) their adolescent’s participation [72,73]. Willis et al. [22] identified that families may benefit from a ‘community of practice’ to support collaboration across contexts and environments. Whilst health professionals commonly focus on current goals—often specific to one life situation, planning for long-term goals, dreams and aspirations across multiple life situations is rarely addressed [74]. Valuing adolescents’ and parents’ voices beyond set time frames of therapy interventions may help enhance motivation, autonomy, and self-determination and promote collaborative, coordinated plans for sustained participation [61,75].

Adolescents and parents highlighted the challenge of ‘being active’ now and ‘staying active’ into the future and the lack of awareness of potentially effective strategies. Achieving sustainable participation in physical activity will require a multi-faceted approach that supports health behaviour change [23,48,76,77,78], grounded in the individual’s subjective experience of involvement, meaning, and mastery, and that considers the wider environment [23,79,80,81,82]. The themes support moving beyond meeting physical activity guidelines to promote the development of quality physical activity experiences, particularly when embarking on a new activity or when pre-existing activity levels have been very low [17,61]. The desired outcome is that adolescents with cerebral palsy and their families identify the different pathways via which they can achieve their personal goals and establish lifelong sustainable participation in physical activity. Using a personalised graphic, like the waka, may help adolescents, their parents, clinicians, and community providers navigate journeys in a meaningful and collaborative way by bringing the right people into the waka at the right time and place before the waka drifts off course. In the waka, new or alternative journeys can be explored, and success will be measured by ongoing attendance and involvement in the physical activities of choice.

Our work aligns well with current theoretical frameworks and models used in health, disability and participation to explain functioning and participation. The Social Ecological Model [46,83] provides an overarching explanation of how systems interrelate. The International Classification of Functioning Disability and Health (ICF) [84] is a bio-psycho-social framework that aims to explain functioning and disability in the presence of a health condition, and the ‘F-words’ of Childhood Disability a family-friendly translation of the ICF [85]. The Family of Participation-Related Constructs (fPRC) [14] is an empirically derived framework based on the ICF that aims to explain participation in more detail, and the Framework for Sustained Participation in Physical Activity by Adolescents with Cerebral Palsy [86] is a synthesis of findings from a study of sustained participation. Table 4 shows how the constructs and conceptual relationships can be aligned to the journey each adolescent takes in their waka towards sustainable participation in physical activity, providing empirical support for varied ways in which sustained participation can be promoted. In addition, the subjective experience of adolescents and their parents matched themes of autonomy, belongingness, challenge, engagement, mastery, and meaning [2,3,87] and many of the strategies developed for the ‘on the ground’ strategy matrix for quality participation experiences [2,88].

### 4.1. Strategies to Enhance Trustworthiness

Trustworthiness was developed using strategies undertaken by the researchers through the 13-month study. This included deep engagement with study participants through regular, individualised, weekly communication in the baseline and HLMP intervention phase and fortnightly thereafter with both the adolescents and their parents. The researcher-participant relationship built a deep understanding of each adolescent’s goals, interests and life demands within their family context. Detailed HLMP intervention notes, an audit trial, a reflective diary, and team discussions enhanced the interpretation of findings. Member checking of themes, graphic development and development of each adolescent’s own story of their journey using the waka was powerful and experienced as meaningful by the adolescents and their families. Fathers and mothers contributed over an extended 13-month period, with 38 of 40 expected interviews conducted. One adolescent (the only female) only attended 2/24 HLMP sessions but completed all follow-up assessments and interviews, adding to the depth and range of experiences shared. Co-design of the graphic with an art teacher who has a son with a disability was a strength and provided a sense of the potential transferability of the findings, as she could reflect on their journey using the data generated from this study.

### 4.2. Limitations

The experiences of only one adolescent female were collected. While her voice highlighted the challenges that can be experienced when barriers to physical activity participation seem insurmountable, gathering a deeper understanding of the perspectives of females with cerebral palsy is needed [89].

### 4.3. Transferability

The concepts and waka metaphor may be generalisable and transferable to a range of people who are considering their participation in physical activity. The concept of enhancing participation in any physical activity is of interest globally: the extent to which these findings transfer across cultures and settings requires further research. The concept of the journey of participation in a waka could be transferable to other disability groups and in different environments. The waka image or analogy could be replaced by an ecologically appropriate model for the adolescent’s environment or preferences, e.g., a sailing ship, wheelchair, or frame runner. The concept of sustained participation could be transferred to other contexts and situations, e.g., wheelchair rugby and athletics in New Zealand and skiing and ice hockey in Canada.

### 4.4. Future Research

Supporting and developing plans for sustaining participation in physical activity into adulthood requires more extensive research, namely, utilising adolescents’ and parents’ strategies, setting goals and following dreams, valuing supporters, and building and expanding support networks. Gaining greater knowledge of what could help, when, where, and how (particularly at critical transition points) may enable daily participation in physical activity to become a reality over the life course for people with cerebral palsy. Research using longitudinal or cross-sectional studies that are co-designed and implemented in collaboration with key stakeholders, including those with lived experience, is needed. Importantly, the inclusion of family units who do and do not participate in physical activity is essential, considering that targeting those who are most inactive could have the greatest benefits. To meet the WHO recommendation for research into physical activity for people with disabilities [12], further longitudinal mixed-methods research on a larger scale is needed. We need to know how to design and deliver interventions at the right dose, in the right place, with the right people, that ensure sustained participation in physical activity into the future for all people with cerebral palsy.

## 5. Conclusions

The ‘Journey to Sustained Participation’ organisational structure, themes and subthemes highlighted the past and present experiences an adolescent living with cerebral palsy could face when active. The Waka graphic tells a story of a journey that could be personalised for adolescents with cerebral palsy participating in physical activity across any context or environment. The journey of sustained participation is dynamic, complex and changeable. Sustaining participation in physical activity is challenging for adolescents living with cerebral palsy and their parents, yet highly valued. For the waka to stay on course, successful journeys need to provide options and opportunities to find the right place, timing, and people. Adolescents and parents must balance the continuum of helpful and hindering experiences and navigate systems to ensure participation in physical activity is part of their life course journey.

## Figures and Tables

**Figure 1 children-10-01533-f001:**
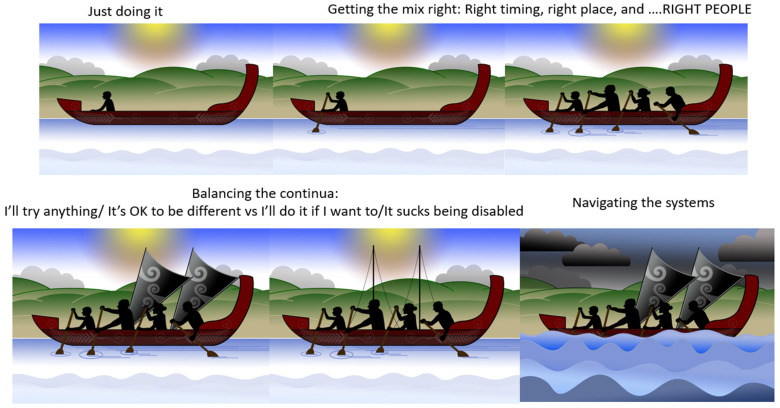
Journey of Sustained Participation. The development of this Taonga (treasure) is endorsed and supported by Toriana Hunt, Kaimahi Māori, Canterbury District Health Board and Manawhenua Ki Waitaha Charitable Trust Trustee.

**Table 1 children-10-01533-t001:** ‘Journey of Sustained Participation’: Themes and Subthemes.

Themes and Subthemes	
**Just doing it**	
- Attendance - Involvement	
**Getting the mix right**	
- Right Place	
- Right Timing	
- Right People	
**Balancing the continua**	
** *I’ll try anything* **	** *I’ll do it if I want to* **
- I am motivated	- I do not want to
- I like it, and it’s my choice	- Do I have a choice?
- It makes me feel good	- It’s not for me
- I will try it if…	- I need help
- I will try it but…	- I do not know if I can
** *It’s OK to be different* **	** *It sucks being disabled* **
- I am as good as me	- Moving towards normal
- I am a para-athlete	- I have fewer options
- I am supported	- I may not be able to when I am older
- I am different	- I am judged
	- I face obstacles
**Navigating the systems**	
- Interpersonal - Organisational - Community - Policy	

**Table 2 children-10-01533-t002:** Experiences influencing and shaping the life course Journey to Sustained Participation in Physical Activity.

Themes and Subthemes: ‘On Track to Sustained Participation’	
**Just doing it**	
Attendance/Involvement *I think two sessions a week. I think that you were really on task, you know they went there, they had a bit of a laugh and a joke, but actually you had a set lesson plan and you just kept it moving. And I think it was a hard hour.* (P) *I think that’s something he wanted to do … he seems to really enjoy it.* (P)	
**Getting the mix right**	
Right place *I’d like for him to have sports that he can go to for enjoyment. Whether it’s mainstream or adapted, but that he’s got options. Even though he’s got the disability. However, I’m not sure that will be our choice to be honest…* (P)	
Right time *I think it’s really benefited… it’s caught him at the right time because he’s in the right place for this now.* (P) *I think he does enough. His academics are really important to him. He actually doesn’t have a whole lot of time. The time he does have I think he needs to be a kid and hang out with his friends. We don’t want every last minute filled up.* (P)	
Right people *It’s just nicer with mum or dad cuz (because) then I don’t need to worry about being left behind or having to catch up. With friends, I’d try to keep up or probably ask them if we can take a rest.* (A) *I think it’s very important when he’s with like-minded people with similar physical abilities. It’s all motivational, he’s definitely in his happy place*. (P) *If I couldn’t find anyone else I could relate to, it would make it more difficult and arduous’.* (A)	
**Balancing the continua**	
** *I’ll try anything* **	** *I’ll do it if I want to* **
I am motivated *I talked to him about intrinsic and extrinsic motivation and I think he’s got that internal motivation for throwing because it holds his interest. If he can keep up that motivation, that focus and that dedication, I think that will serve him well.* (P)	I do not want to *I just can’t seem to get him to do his home exercises. So, I think it’s good to be doing some programme that kind of forces him to do it. You can see that he’s doing it properly and give them advice and he will take it from you because what do I know?* (P)
I like it and it’s my choice *Basketball’s just a fun sport to play, that I enjoy and actually I’m kind of good at. Like soccer, I’m terrible. I can’t kick very hard compared to my brother*. (A)	Do I have a choice? *Swimming is not his choice. Because I don’t want him to drown. It’s a life skill rather than a choice*. (P)
It makes me feel good *It makes me feel better. Healthier in my body. I feel good that I’m doing it*. (A)	It’s not for me *He’s not a physical person. He’s an indoor Lego, intellectual person. Some people get a buzz out of exerting themselves and others don’t. He’s the type that doesn’t*. (P)
I will try it if… *I think because I need people to do it with me so I can base myself off them and compete with them. Using my competitive side to motivate me*. (A)	I need help *We admit he is a teenage boy and without a little swift kick in the bum, he would quite happily sit on his bum all day. There’s a lot of motivation from either myself, his mum or even his grandmother…We appreciate he’s a teenage boy, and he needs a little bit of direction*. (P)
I will try it but… *I’m quite a bit tired sometimes doing stuff. I just want to do it. However, my body says I actually can’t because I’m so tired…I get very, very sore muscles. … I would take all my sores and just throw it away and just try to keep going… However, sometimes it overwhelms me and just goes on. And I just can’t do anything*. (A)	I do not know if I can *I don’t think I’m really that good enough to be in a basketball team. And they look really good compared to me. I’ve never really thought of actually playing sports*. (A)
** *It’s OK to be different* **	** *It sucks being disabled* **
I am as good as me *There are people at school, which are just like, “oh, you can’t do sport, you’re disabled”. However, it’s about teaching them, that, while you might not view me as very good at sport, if you look at my results in a para perspective, I’m actually doing very well*. (A)	Moving towards ‘normal’ *I think it’s important that we do things to keep him active so that when his friends want to do things, he can keep up with them. Because that has been a problem in the past. You know, his friends are not disabled. He has got a bit upset about it, that he can’t do what they do*. (P)
I am a para-athlete *You have to be patient and you can’t get everything right the first time…I have to work harder…I want to win…I’d like to get good enough to get internationally classified or compete internationally*. (A)	I have fewer options *In cricket we said I might not be able to play hard ball. I might have to have a special thing where I get to run early but I just want to be like everyone else and running at the normal time. I wouldn’t enjoy it at all*. (A)
I am supported *Just encouraging her. Know that it’s OK. Everyone is different, not everyone’s the same. I always tell my kids, who cares what they think?* (P)	I may not be able to when I am older *Hopefully I’d still be running but probably not … It’s quite important to keep up running because otherwise my muscles aren’t going to be functioning as well in the future. It would be harder to walk”*. (A)
I am different *The girls will often bike and sometimes he’ll get on his three-wheeler bike*. (P)	I am judged *There’s activity she would love to do but she just doesn’t do it. Because of her confidence and worrying about what other people will think. It’s a complex thing. You know, being her age, 14, and having other kids looking in on her*. (P)
	I face obstacles *Trying to find her something to do is really difficult. I don’t know the pain that she’s feeling every day when she gets home after school, … so I can’t really force her to go and do something*. (P)
**Navigating the systems**	
Interpersonal *For us as a family it’s been really good that my son had something that he has been participating in for himself and been achieving within that. I mean we all love him succeeding and even his brother does. We’ll be able to do more. We’re always looking for activities that all of us can enjoy at the same time*. (P) Organisational *We were all talking about why we haven’t known any of this sort of stuff from the HLMP earlier*. (P) *When we have relievers in [substitute teachers], that’s the big thing, they don’t understand my disability fully*. (A) Community *Everything’s a minefield, you’ve got to do this, this, this and this. You almost need a guide to guide your way through these things because it’s not easy if you’re just dipping your foot in for the first time*. (P) Policy *With COVID lockdown, athletics training ended. Missed athletics sports, chance to compete at big events. Missed training leading up to Halberg games. Missed my birthday party which was go-carting. Can’t go swimming. Missed biking, chain is broken and can’t buy new part. Can’t walk in the hills with my walking group. Not walking to and from school bus as no school*. (A) *He coped with COVID lockdown well. He didn’t like that he couldn’t get out and about to do the activities or see people that he wanted to see. However, he was able to do more physical activity because he wasn’t at school and didn’t have rigid hours or yeah and wasn’t as tired. And COVID gave him the freedom to because there was less traffic around, that we were happier for him to go walking by himself around the local area (which he had never done or wanted to do before). And, yeah, that was really positive for him and made him feel more confident and able…* (P)	

**Table 4 children-10-01533-t004:** ‘Journey to Sustained Participation’ Aligned to Theoretical Frameworks and Models.

Themes	ICF	F-Words	fPRC	Social Ecological Model	Framework for Sustained Participation
‘Just doing it’ 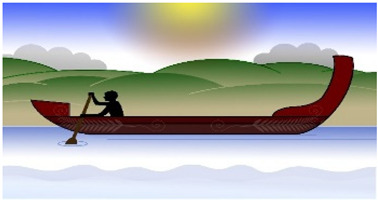	Participation	Friendship Fun	Participation: Attendance Involvement	Intrapersonal	Getting started
‘Getting the mix right’ 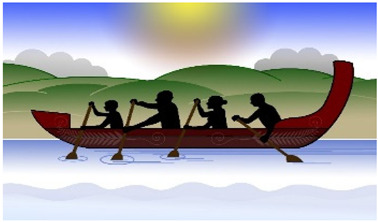	Contextual factors	Family factors	Contextual factors	Intrapersonal Interpersonal	Sense of belonging The coach is important Being passionate
‘Balancing the continua’ 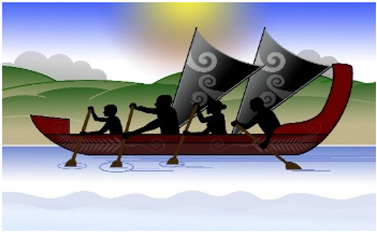	Body function and structure Activity Participation Personal factors	Fitness Function Friendship Fun	Preferences Activity competence Sense of self Transactional relations between constructs	Intrapersonal	Wanting to succeed Being passionate Sense of belonging
‘Navigating the systems’ 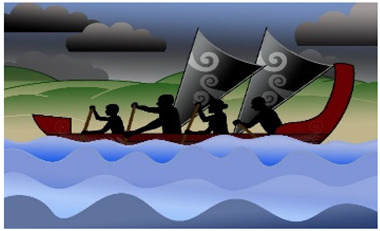	Environmental and contextual factors	Family factors Future	Contextual and environmental factors	Interpersonal Organisational Community Public policy	The coach is important Endorsement of continue Endorsement to support

Notes: ICF, International Classification of Functioning, Health and Disabilities Framework [84]—a bio-psycho-social model of disability; F Words in Childhood Disability [85]—a family-friendly translation of the ICF; fPRC —Family of Participation Related Constructs [14]—a more detailed participation-focused framework based on the ICF; Social Ecological Model [46,75,83]—Intrapersonal/Individual—knowledge, attitudes, beliefs, personality traits; Interpersonal—family, friends, social networks; Organisational—therapy services, schools, disability groups; Community—social networks, relationships between organisations, design, access, connectedness, spaces; Policy—local, national and international laws and policy; A Framework for Sustained Participation in Physical Activity by Adolescents with Cerebral Palsy [86]—a synthesis of findings of a study focused on sustaining physical activity participation.

## Data Availability

De-identified data from this study may be made available on request to the corresponding author if appropriate ethical approval for the subsequent use of the data is in place.

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
