# Peer review of "The Journey to Sustainable Participation in Physical Activity for Adolescents Living with Cerebral Palsy"

_children, 2023, doi:10.3390/children10091533_

Round 1
Reviewer 1 Report
Purpose of this interpretive descriptive study is to understand adolescents diagnosed with CP and their parent’s perspective on being active.
This is a great manuscript covering very important complex topic.
Please consider the minor suggestions:
Lines 125-133.
Please provide a brief description of participation level of adolescents at the beginning of the study. From some of the subjective comments is appears some of the adolescents were already active in sports. I believe providing a very brief description of their physical activity level prior to participating in the study would be beneficial to know.
Lines 150-151.
Please clarify the sentence “These schedules were also piloted prior to delivery with the same nine children as the previous pilot.”
Line 192. Please provide a one sentence describing ‘the waka’.
Lines 211-212. Please clarify or expound what ‘additional information’ was provided.
Lines 485-500 are the beginning of a great summary!
Author Response
Thank you very much for taking the time to review this manuscript. Please find the detailed responses below and the corresponding revisions/corrections highlighted/in track changes in the re-submitted files.
- Please provide a brief description of participation level of adolescents at the beginning of the study. From some of the subjective comments is appears some of the adolescents were already active in sports. I believe providing a very brief description of their physical activity level prior to participating in the study would be beneficial to know. Thank you I have added the following description as suggested: Five of the adolescents were involved in at least one formal, organized physical activity prior to the programme e.g., karate, athletics, swimming lessons, cricket, football. No adolescent received any form of physiotherapy during the intervention or follow up period
- Please clarify the sentence “These schedules were also piloted prior to delivery with the same nine children as the previous pilot.” Thank you - added: Interview schedules two and three were also piloted prior to delivery, with the same nine children as in interview one, to ensure age appropriate language and meaning.
- Please provide a one sentence describing ‘the waka’.Thank you.I have added a sentence. A waka is a traditional Māori watercraft like a canoe, used by Māori (indigenous people of Aotearoa, New Zealand) to journey across the Pacific Ocean to discover Aotearoa, New Zealand. Waka are wooden, ranging in size and specification (single and double hull). The ornamental carvings adorning the waka are deeply culturally significant.
- Please clarify or expound what ‘additional information’ was provided. Added to clarify:
Following interviews, additional information was provided from the adolescent-parent dyad via email, telephone or text e.g., further reflections or specific examples to their interview responses that they had not considered during the interview.
Reviewer 2 Report
This is an interesting report on an interpretive description qualitative study to explore the journey to sustainable participation in physical activity for adolescent with cerebral palsy.
The presentations of the results are long and narrative, but I think this is acceptable due to the nature of the research method.
I think the metaphor of the waka is interesting and contribute to recognize the whole message.
Minor issue
Line 601. Font sizes are not consistent.
Author Response
Thank you very much for taking the time to review this manuscript. Please find the detailed responses below and the corresponding revisions/corrections highlighted/in track changes in the re-submitted files.
I have reviewed font sizes and corrected throughout the paper
Reviewer 3 Report
Thank you for opportunity to review.
It was stated that this study was conducted as part of another study.
This interpretive description study was completed as part of a larger non-randomised, concurrent, single subject research design (SSRD) study investigating the effect of a HLMP on participation in physical activity (Kilgour et al, in press).
Despite the recent application of an integrated research method that conducts both quantitative and qualitative research at once, it is necessary to present a valid reason for dividing the research.
Please provide more detailed information such as mother’s socioeconomic characteristics so that we can understand the study participants.
There are a few editing errors in the manuscript.
Suggest revising the title to “The journey to sustainable participation in physical activity for adolescents with cerebral palsy and their parents.
Author Response
Thank you very much for taking the time to review this manuscript. Please find the detailed responses below and the corresponding revisions/corrections highlighted/in track changes in the re-submitted files.
- This interpretive description study was completed as part of a larger non-randomised, concurrent, single subject research design (SSRD) study investigating the effect of a HLMP on participation in physical activity (Kilgour et al, in press).Despite the recent application of an integrated research method that conducts both quantitative and qualitative research at once, it is necessary to present a valid reason for dividing the research.Thank you for your comment.
The design of the overall study was to conduct concurrent but not mixed quantitative and qualitative methods to achieve our overall aim of evaluating the effects of an intervention and deepening our understanding of the experience of physical activity participation. We have inserted a sentence in the methods that states the overarching method: “The study was concurrent to, but not mixed, with, the SSRD.”
2. Please provide more detailed information such as mother’s socioeconomic characteristics so that we can understand the study participants. Thank you. We did not collect demographic information for the parents and therefore cannot share these details. We have also been recommended by PhD reviewers to de-identify all adolescent details further based on the small sample size and small population pool in which they were drawn.
3.There are a few editing errors in the manuscript. Thank you - I have reviewed and made corrections
4. Suggest revising the title to “The journey to sustainable participation in physical activity for adolescents with cerebral palsy and their parents. Thank you -
The team gave consideration to including the ‘parents’ in the title as parents certainly support the journey. However, we decided the journey itself that is in focus, was that of the adolescent when participating. We have not changed the title.